# p53 and Myofibroblast Apoptosis in Organ Fibrosis

**DOI:** 10.3390/ijms24076737

**Published:** 2023-04-04

**Authors:** Kealan McElhinney, Mustapha Irnaten, Colm O’Brien

**Affiliations:** UCD Clinical Research Centre, Mater Misericordiae University Hospital, D07 R2WY Dublin, Ireland

**Keywords:** fibrosis, p53, apoptosis, myofibroblast, extracellular matrix, glaucoma

## Abstract

Organ fibrosis represents a dysregulated, maladaptive wound repair response that results in progressive disruption of normal tissue architecture leading to detrimental deterioration in physiological function, and significant morbidity/mortality. Fibrosis is thought to contribute to nearly 50% of all deaths in the Western world with current treatment modalities effective in slowing disease progression but not effective in restoring organ function or reversing fibrotic changes. When physiological wound repair is complete, myofibroblasts are programmed to undergo cell death and self-clearance, however, in fibrosis there is a characteristic absence of myofibroblast apoptosis. It has been shown that in fibrosis, myofibroblasts adopt an apoptotic-resistant, highly proliferative phenotype leading to persistent myofibroblast activation and perpetuation of the fibrotic disease process. Recently, this pathological adaptation has been linked to dysregulated expression of tumour suppressor gene p53. In this review, we discuss p53 dysregulation and apoptotic failure in myofibroblasts and demonstrate its consistent link to fibrotic disease development in all types of organ fibrosis. An enhanced understanding of the role of p53 dysregulation and myofibroblast apoptosis may aid in future novel therapeutic and/or diagnostic strategies in organ fibrosis.

## 1. Introduction

Fibrosis is characterized by pathological exuberant deposition of extracellular matrix (ECM) that leads to distortion of tissue architecture and loss of cellular homeostasis. In simplistic terms, fibrosis is uncontrolled wound healing and scar formation [1,2]. Fibrosis poses diagnostic and therapeutic challenges in ophthalmology for conditions such as glaucoma and macular degeneration [1]. Extraocular fibrosis is evident in conditions such as cardiac fibrosis, pulmonary fibrosis, liver cirrhosis, progressive kidney disease, and systemic sclerosis [3]. Furthermore, abnormal fibrotic processes can also have implications for cancer metastasis and graft rejection [4].

It is estimated that in Western developed countries almost 50% of all deaths may be attributed to fibrosis [5]. Currently, the restoration of normal tissue through fibrosis resolution is poorly understood, and with few recognized fibrotic therapies available, it is paramount that further research is implemented in fibrosis.

## 2. Physiological Wound Repair

To effectively describe fibrotic disease, it is essential to understand the physiological wound repair response system which involves intricate, coordinated regulation of numerous cells both temporally and spatially. A range of cells are involved—epithelial cells, vascular endothelium, mesenchymal cells, and inflammatory cells—together these cells interplay as part of an adaptive ECM regulated by soluble mediators, and coagulation factors. plasma proteins, and biomechanical cues [6]. When successful, the wound repair response re-establishes an intact epithelial barrier and clearance of ECM components. This is achieved through overlapping activation of the coagulation, inflammatory, proliferative, and remodeling stages [7,8].

Firstly, tissue injury will initiate the anti-fibrinolytic-coagulation cascade that will result in platelet influx to form a fibrin clot abundant with fibronectin [2]. Platelets play an important role with their aggregation leading to hemostasis and their degranulation releasing cytokines and growth factors such as transforming growth factor-beta (TGF-β). Next, an inflammatory stage begins with neutrophils and monocytes/macrophages recruitment. Activated neutrophils and macrophages phagocytose damaged tissue, efferocytose dead cells, protect against invasive organisms, and amplify the acute inflammatory response [9,10].

In the proliferative stage, epithelial and endothelial cells migrate to re-establish barrier function with accompanying angiogenesis. Next, fibroblasts are recruited, proliferate, and migrate until they are the predominant cell type in the wound bed [11]. Fibroblasts are a heterogeneous, collagen-producing cell group that are ubiquitous in connective tissue and responsible for physiological ECM homeostasis [12]. Fibroblasts can be derived from different progenitor populations such as endothelial cells, pericytes, epithelial cells, mesenchymal stem cells, pre-adipocytes, and adipocytes [13,14,15,16,17,18]. In the remodeling phase of wound repair response, fibroblasts are activated through the influence of biochemical and biomechanical stress factors [19] and differentiate into myofibroblasts which are professional repair cells [11,13]. 

Myofibroblasts are smooth muscle-like contraction cells that express organised stress fibres consisting of myosin filaments and alpha-smooth muscle actin (α-SMA) [20,21], that promote wound contraction and help oppose wound edges [22,23,24]. Additionally, myofibroblasts secrete ECM proteins [25], degrade ECM by matrix metalloproteinases [26], and organize/remodel ECM fibres to increase mechanical stability [27]. Myofibroblasts orchestrate wound repair leading to the production and maturation of a collagen-rich scar and tissue integrity restoration [10,23].

## 3. Dysregulated Wound Repair

In fibrosis, the physiological wound repair response becomes imbalanced, leading to a dysregulated wound repair response [28]. Fibrosis is characterized by a vicious cycle of recurrent injury to the epithelium/endothelium [29] resulting in the activation and accumulation of myofibroblasts [2,30]. Persistent tissue injury or irritants may originate from a plethora of sources such as oxidative stress, allergens, hypoxia, recurrent inflammation, trauma, drugs, toxins, mechanical stress, idiopathic, or unknown [31]. The presence of a persistent irritant is common in most fibrotic conditions including pulmonary fibrosis and renal fibrosis [32]. Furthermore, pro-fibrotic biochemical mediators and stiffened ECM biomechanical stimuli further potentiate myofibroblasts activation [33,34,35].

In this setting, chronic inflammation, injury, and repair can lead to excessive production of ECM (collagen, fibronectin, proteoglycans, and hyaluronic acid) [32]. Pathological levels of ECM lead to progressive remodeling and loss of normal tissue architecture [20,36], detrimentally affecting normal tissue’s ability to carry out its physiological duties [13]. Examples of this detrimental effect on physiological function include the pulmonary alveolus in idiopathic pulmonary fibrosis [37], the renal tubular epithelium in obstructive nephropathy [38], the normal hepatic parenchyma in liver fibrosis [39], cardiac cardiomyocytes in cardiac fibrosis [40], and keloid scarring of dermal wounds [41].

## 4. Apoptosis

Following the completion of physiological wound repair, myofibroblasts are terminated through apoptosis [42,43]. Apoptosis is a complex, coordinated physiological process that results in programmed cell death [44,45]. Cellular apoptosis eliminates damaged or infected cells, regulates organ size and function, and prevents the propagation of uncontrolled cancerous cells [46,47]. Apoptosis is important in embryological development, for example, apoptosis in cells between the fingers of an embryo results in the interdigit spaces [48]. In ocular development, ganglion cells that leave the retina but do not synapse the lateral geniculate nucleus undergo physiological apoptosis [49]. 

Apoptosis involves sequential cell shrinkage, chromatin condensation (pyknosis), nuclear fragmentation (karyorrhexis), plasma membrane blebbing/protrusion, followed by cell fragmentation (budding) [50]. Resultant apoptotic bodies undergo phagocytosis by macrophages (efferocytosis). This process prevents the release of intracellular components such as inflammatory factors—described as “clean” cell death [48]. In contrast, necrosis involves cell swelling, membrane rupture, and release of intracellular components, inciting an inflammatory response with damage to surrounding structures [50,51]. Apoptosis can be initiated by two interconnected molecular signaling cascades: “extrinsic” (death receptor) and “intrinsic” (mitochondrial) pathways [50]. These pathways converge at the level of the caspases (Cysteine aspartic acid proteases) to initiate the “execution phase” of apoptosis [52,53]. Programmed cell death is subsequently triggered—this cannot be halted once initiated [54].

When apoptosis is instigated by cellular stresses from inside the cell, it will initiate the intrinsic or mitochondrial pathway. Intracellular stimuli include oxidative stress (hypoxia), damaged DNA, activation of an oncogene, radiation, growth factor deprivation, cytoskeletal disruption, or accumulation of unfolded proteins [45]. The “intrinsic” (mitochondrial) pathway is primarily regulated by the B cell lymphoma 2 (BCL-2) family of proteins through dynamic interactions at the mitochondrial outer membrane [44,55]. Apoptotic stimuli will cause pro-apoptotic BCL-2 proteins to be transcriptionally or post-transcriptionally upregulated. When pro-apoptotic BCL-2 proteins outweigh anti-apoptotic BCL-2 proteins, apoptosis proceeds through the pore formation in the mitochondrial membrane (mitochondrial apoptosis-induced channels) and the induction of mitochondrial outer membrane permeabilization (MOMP) [56].

BCL-2 family member proteins are divided into effector, activator, anti-apoptotic and sensitizer proteins based on structure, function, and presence of one or more of the four BCL-2 homology (BH) domains [44,45]—BH1, BH2, BH3, and BH4 [57]. Multi-domain effector proteins BCL-2 homologous antagonist/killer (BAK) and BCL-associated X protein (BAX) initiate apoptosis via MOMP [58]. These effectors are regulated by BH3-only activator proteins, such as BH3-interacting domain death agonist (BID), a p53-upregulated modulator of apoptosis (PUMA), and BCL-2-like protein 11 (BCL2L11; or BIM) which bind effectors and initiate MOMP [59]. This interaction may be inhibited by anti-apoptotic proteins like BCL-2, BCL-W, BCL-XL, induced myeloid leukemia cell differentiation protein MCL1 (MCL-1) and BCL-2-related protein A1 (BCL2A1; or BFL1) that inhibit effectors/activators to halt MOMP [55]. Despite the presence of anti-apoptotic proteins, MOMP may still commence in cells if there is an elevated expression of BH3-only sensitizer proteins. Sensitizers include BCL-2-interacting killer (BIK), BCL-2-associated death promoter (BAD), activator of apoptosis harakiri (HRK), phorbol-12-myristate-13-acetate-induced protein 1 (PMAIP1; or NOXA), PUMA (also an activator), and BCL-2 modifying factor (BMF) may promote apoptosis by indirectly inhibiting anti-apoptotic proteins and thusly enabling activators/effectors MOMP initiation [55] (Figure 1).

MOMP results in mitochondrial swelling, rupture of the outer mitochondrial membrane, and cytochrome c release into the cytoplasm [2,11]. Cytochrome c activates apoptotic protease-activating-factor-1 (APAF-1) to form an apoptosome. The apoptosome is responsible for the transformation of procaspase-9 to activated caspase-9 which can subsequently directly stimulate “executioner caspases” (caspase-3 and caspase-7) commencing organised cellular destruction [54,56,60]. The “execution phase” of apoptosis is regulated by a pro-survival inhibitor of apoptosis proteins (IAPs) (X-linked IAP (XIAP), cellular IAP-1, cellular IAP-2, survivin, and livin) [48] which have an inhibitory effect on caspase activation [56]. However, these may be neutralized by pro-apoptotic secondary mitochondria-derived-activator-of-caspases (SMAC/Diablo) and high-temperature-requirement-serine-protease (HtrA2/OMI) [55,56,61].

When the commencement of apoptosis originates from outside the cell, it is part of the extrinsic or death receptor pathway. Extracellular cues are delivered to transmembrane death receptors in the form of ligands that instigate intracellular signaling cascades to culminate in cell death [62]. These receptors belong within the tumour necrosis factor (TNF) gene superfamily [56] and include the first apoptosis signal (Fas), death receptor-4 (DR4), and death receptor-5 (DR5). Ligands include TNF, TNF ligand superfamily-member 10 (TNFSF10; or TRAIL), and Fas ligand (Fas-L) [50] (Figure 1).

Each transmembrane death receptor furnishes an intracellular death effector domain. Upon ligand binding, death domains will recruit and bind adapter proteins TNFR/Fas-associated-death-domain (TRADD/FADD) with procaspase-8 and form a death-inducing signaling complex (DISC). DISC are responsible for caspase-8 activation which can directly initiate apoptosis through capase-3 and caspase-7 activation [54].

Extrinsic/intrinsic apoptotic pathways do not act as independent, parallel pathways that simply converge on common caspase machinery [56]. Cross-activation between pathways occurs through the activator protein BID [63]. The extrinsic pathway activates BID through caspase-8 mediated proteolytic cleavage [64] which can then trigger the effector proteins through BAK and BAX [65].

Extrinsic/intrinsic apoptotic pathways converge at the level of the caspases. Caspases are cysteinyl aspartate-specific proteases that are essential in maintaining homeostasis. Caspases also play a role in inflammation (e.g., caspase-1, -4, -5, -13, and -14) [54,56]. Apoptotic caspases are subdivided into “initiator caspases” (e.g., caspase-2, -8, -9, and -10), that initiate apoptosis, and “executioner caspases” (caspase-3, -6, and -7) that cleave cellular elements in cell death [66]. Downstream, caspases are responsible for the cleavage of protein kinases, DNA repair proteins, and cytoskeletal proteins. They also interact with the cytoskeleton and cell cycle signaling pathways contributing to apoptotic morphological alterations [67]. Cell death is subsequently triggered, and cannot be halted once initiated [54].

## 5. p53

As noted above, when physiological wound repair is complete, myofibroblasts undergo programmed cell death and self-clearance [42,43]. However, in fibrosis, there is a characteristic absence of myofibroblast apoptosis [13,60,68]. Key research groups in fibrosis have shown that myofibroblasts responsible for systemic fibrotic disease development adopt an apoptotic-resistant, highly proliferative phenotype [2,11]. This results in the persistence of activated myofibroblasts that perpetuate the pathological fibrotic disease process [69] and excessive ECM synthesis, deposition, and remodeling [23,70]. Significantly, this pathological adaptation has been linked to dysregulated expression of tumour suppressor gene p53 [71].

Tumour suppressor gene p53 was discovered over 40 years ago [72] and has been widely studied in cancer formation and/or progression [73]. p53 is located on human chromosome 17 consists of 393 amino acids and is named after its 53 kDa relative molecular mass [73]. p53 is a potent transcription factor [74] that is activated in response to environmental insults and diverse stresses, and is responsible for the induction of cell-cycle arrest, apoptosis, and/or senescence [75]. p53’s primary function is to prevent the emergence of transformed cells with genetic instabilities and it is therefore essential in preventing cancer onset and development [76], earning its title “Guardian of the Genome” [77]. 

p53 is important in cell-cycle regulation through its role as a checkpoint protein, inducing cell cycle arrest at the G1-S and G2-M checkpoints [78,79]. At the G1-S checkpoint, cell-cycle arrest is p53 dependent. Typically, cellular levels of p53 are low but DNA damage can result in prompt induction and activation of p53 [80]. DNA damage is recognized by protein kinases (ataxia-telangiectasia-mutated (ATM), ataxia-and-rad3-related (ATR)) that activate p53 by phosphorylation [81]. p53 stimulates p21 transcription [82]—a cyclin-dependent kinases (CDK) inhibitor of G1-CDK-cyclin complexes (CDK2-cyclin-E, CDK4-cyclin-D), and CDK1-cyclin-B complexes [83,84]. Inducing p21, therefore, arrests the cell-cycle at the G1 phase preventing the replication of damaged DNA [83] (Figure 2). 

In a similar fashion, p53-dependent DNA damage response increases transcription of p21 and 14-3-3 sigma (14-3-3 σ). p21 inhibits CDK1-cyclin B complexes while 14-3-3 σ actively excludes cyclin B from the nucleus. Both result in cell cycle arrest at the G2-M checkpoint [85]. p53 also mediates the dissociation of CDK1-cyclin B complexes by induction of GADD45 (growth arrest and DNA damage-inducible gene [78]. 

When cell-cycle arrest initiated by p53 is insufficient for DNA repair, p53 will initiate apoptosis [86]. p53 regulates the transcription of a variety of genes integral to apoptotic signaling through extrinsic and intrinsic pathways [87]. p53-induced targets include those involved in the extrinsic pathway (Fas, DR4, DR5), intrinsic pathway (BCL-2 family member proteins PUMA, NOXA, BAD, BAX, BIM, BAK), and execution factors (APAF-1, caspase 6) [84,88]. p53 also enables apoptosis by downregulation of anti-apoptotic BCL-2 family member proteins BCL-2, MCL-1 [89] (Figure 2). 

p53 also induces apoptosis via transcription-independent means [90]. p53 has been shown to migrate to mitochondria and directly induce MOMP through binding to anti-apoptotic proteins BCL-2 and BCL-XL, promoting the release and activation of effector proteins BAK and BAX [84]. Direct activation of BAK by p53 has also been appreciated [91]. p53 can even directly induce MOMP in ischemic models, independent of BAX/BAK but dependent on cyclophilin D [92].

One of the regions of p53 that is believed to be required for its ability to induce apoptosis is its proline-rich domain [93]. Within this domain, there is a common single-nucleotide polymorphism (SNP) at codon 72 encoding either an arginine (R72) or a proline (P72) residue [93]. Interestingly, this variance in polymorphic form results in a marked alteration in the biochemical structure and function of p53 [94]. In-vitro studies have shown that p53 with expressions of the P72 allele demonstrate an increased ability to induce senescence and cell-cycle arrest [95] through increased transactivation of p21/Waf-1, which blocks CDK activity leading to growth arrest at the G1 phase of the cell-cycle [96]. p53 with expressions of the R72 allele exhibit an increased ability to induce apoptosis [97] thought to be related to elevated mitochondrial localization and stimulation of pro-apoptotic BCL-2 family member protein BAK [98].

In most, if not all, human cancers, inactivation of anti-proliferative and pro-apoptotic p53 disrupts its ability to suppress carcinogenesis, thus transforming the “Guardian of the Genome” into a “Rebel Angel” [99]. As noted above, p53 inactivation has recently been linked to fibrotic disease development [71]. Therefore, this review aims to summarize what is currently known about p53 and apoptotic failure in myofibroblasts and the role they play in organ fibrosis. 

## 6. Lung Fibrosis

Fibrosis of the lung is characterized by excessive collagen deposition with fibrotic foci containing endothelial cells [100]. Pulmonary fibrosis is caused by a range of etiologies that include idiopathic pulmonary fibrosis (IPF), scleroderma, infectious, sarcoidosis, radiation, or toxic causes [28]. 

Idiopathic pulmonary fibrosis (IPF), the most common form of idiopathic interstitial pneumonia, is characterized by progressive lung scarring and disruption of physiological tissue architecture, resulting in respiratory failure and death [3]. IPF has a poor prognosis, with median survival reported to be 2 to 3 years [101]. Currently, the only approved fibrotic therapies for IPF include pirfenidone and nintedanib [102]. These medications can slow but cannot halt IPF progression or reverse fibrotic changes [31]. Experimental models for pulmonary fibrosis include the murine models of intratracheal administration of bleomycin, amiodarone, or asbestos [28,103]. Other less commonly used models involve the administration of amiodarone, or asbestos [104].

Following insult to lung tissues, damage to type-II alveolar epithelial cells (AECs) initiates wound repair response [99]. AECs are found in close proximity to pulmonary fibroblasts [105] and together they participate in a reciprocal activating relationship [106] through the release of pro-fibrotic cytokines such as TGF-β1, platelet-derived growth factor (PDGF), and TNF-α [107,108,109,110]. Activated pulmonary fibroblasts (myofibroblasts) are then responsible for the deposition of collagen [106,111]. However, pulmonary fibrosis can develop in response to various stimuli, for example, recurrent AEC injury or chronic inflammation [112,113]. Continued sustained myofibroblast activation results in excessive ECM deposition and pulmonary fibrosis [30,114,115]. 

Pulmonary fibroblasts are recruited from resident fibroblasts and circulating (bone marrow-derived) fibrocytes [97]. Transdifferentiation also occurs through pulmonary epithelial-mesenchymal transformation (EMT) to supplement the fibroblast population [99,100,116]. Recent studies have also shown that pulmonary myofibroblasts are also recruited from pericytes and from mesothelial to mesenchymal transition (MMT) [117,118,119,120,121].

Pulmonary fibroblasts and their activated myofibroblast form have been extensively studied and have been found to drive IPF disease development. Apoptosis-resistance is believed to play an essential role in fibrogenesis in IPF patients with pulmonary myofibroblast persistence leading to excessive levels of ECM deposition, persistent tissue tension/contraction, and the formation of a pathological scar [106]. Studies have consistently shown decreased levels of apoptosis in pulmonary fibroblasts to be directly linked to pulmonary fibrogenesis [60,122,123,124,125]. Simultaneously, excessive apoptosis of juxta-positional AEC has also been shown to facilitate pulmonary fibrosis development [126] as part of what has been described as the “apoptosis paradox” [127,128]. 

Myofibroblast survival is a key determining factor in pulmonary fibrotic disease progression [129] and is made possible through pro-survival mechanisms such as the release of regulatory cytokines. TGF-β is a multifunctional cytokine involved in the regulation of inflammation, wound healing, and ECM production [130,131]. TGF-β1 mediates myofibroblast activation, the pathophysiology of fibrosis and mediates myofibroblast activation [13,132]. TGF-β1 diminishes myofibroblast susceptibility to apoptosis by β1 integrin [133,134]. Activation of pro-survival protein kinase pathways involving phosphoinositide 3-kinases/Ak strain transforming (PI3K/AKT) and focal adhesion kinase (FAK) [135,136]. Furthermore, it directly interferes with apoptotic signaling by regulating BCL-2 family member protein expression [30,59], upregulating IAPs survivin and XIAP [137,138], and suppressing Fas (CD-95) expression [139]. In pulmonary fibrosis, many of these signaling pathways are also activated by vasoactive peptide endothelin-1 (ET1) [135,140] and lysophosphatidic acid [141].

Interestingly, Hinz et al. have recently proposed that fibrotic apoptotic-resistant pulmonary myofibroblasts are simultaneously poised to self-destruct [11] through increased mitochondrial pro-apoptotic priming [59,142]. This means myofibroblasts are prevented from crossing the apoptotic threshold through increased expression of anti-apoptotic BCL-2 family member proteins relative to pro-apoptotic counterparts [143,144,145]. Pro-apoptotic BH3-only proteins can be induced by cytotoxic stress signals, thereby increasing mitochondrial priming [55,145]. When mitochondrial priming is high enough to cross the apoptotic threshold, MOMP, and subsequent apoptosis will occur [11]. 

Importantly, cells can still survive if a pro-survival mechanism is activated. Typically, cells with high mitochondrial priming upregulate anti-apoptotic proteins that sequester pro-apoptotic BH3-only proteins to prevent MOMP [144,146,147]. In myofibroblasts, upregulation of anti-apoptotic BCL-2 proteins (e.g., BCL-XL) enables cell survival despite being primed for death [59]. Hence, these cells are dependent on anti-apoptotic proteins for survival, and the inhibition of these proteins (e.g., BH3 mimetics) can rapidly induce apoptosis in such cells [147]. 

Targeting myofibroblast apoptosis is a growing therapeutic strategy aimed at reversing fibrosis [59,148,149]. Inhibition of pro-survival IAP family proteins reduces bleomycin-induced lung fibrosis [150], as does upregulation of Fas expression induced by quercetin [151] or administration of TNF-α [152]. Targeted inhibition of the pro-survival BCL-2 family proteins using BH3-mimetics was shown to promote myofibroblast apoptosis and reverse tissue fibrosis in murine disease models [59,148,149]. Importantly, only cells with mitochondrial apoptotic priming are sensitive to these drugs [44,45]. Preliminary experiments have shown that BH3 mimetic ABT-263 (navitoclax) binds and inhibits BCL-2, BCL-W, and BCL-XL to reverse established fibrosis in preclinical models of lung fibrosis [59,153]. Further clinical studies are needed regarding BH3 mimetic drugs in human fibrosis.

The relationship between p53 and pulmonary fibrosis was first investigated following evidence of p53 overexpression by immunostaining in bronchoepithelial cells in patients with IPF [154]. This finding led to Hojo et al. to investigate further via fluorescence-based single-strand conformation polymorphism, cloning–sequencing, and immunostaining. This group demonstrated that bronchoepithelial cells in IPF patients frequently had heterogeneous point mutations of the p53 that predominanaffectedcted the central area of the gene [155]. 

Kuwano et al. in 1997 demonstrated p53 over-expression in the epithelial layers of lung tissues obtained from IPF patients but not in control tissues [156]. Furthermore, this study showed that p53 and p21 upregulation was associated with chronic DNA damage, cell-cycle arrest, and apoptosis in IPF tissues (compared to normal control)—this was quantified through immunohistochemistry and Terminal deoxynucleotidyl transferase dUTP nick end labelling assay [156]. This finding was further extrapolated by Lok et al. 2001 who demonstrated increased wild-type p53 expression in epithelial lung tissues from IPF patients [157]. Murine pulmonary fibrosis models utilizing intratracheal administration of bleomycin have also demonstrated significant p53 over-expression in epithelial cells from fibrotic samples compared to normal controls [158]. 

p53 expression in pulmonary fibroblasts was extensively researched by Nagaraja et al. in 2018 who showed that p53 expression is reduced in pulmonary fibroblasts in IPF patients and murine bleomycin-induced pulmonary fibrosis compared with its expression in normal pulmonary fibroblasts [71]. Furthermore, this group also demonstrated that inhibition of baseline p53 in control pulmonary fibroblasts increased profibrogenic protein expression, and restoring p53 to fibrotic pulmonary fibroblasts reduced profibrotic signaling [71]. This suggests that the loss of basal p53 in pulmonary fibroblasts permits the production of excessive ECM proteins [159,160].

Together these findings may be in keeping with the theory of the “apoptosis paradox” with p53 over-expression potentiating continuous AEC apoptosis [161,162,163] that are replaced with pulmonary myofibroblasts that adopt an apoptosis-resistant phenotype aided by p53 under-expression leading to the destruction of lung architecture, excessive ECM deposition and progressive loss of lung function [159,164,165,166].

## 7. Liver Fibrosis

The extensive scientific effort has enabled a greater understanding of the pathophysiological mechanisms causing liver fibrosis—partly because the liver is the only mammalian organ known to regenerate after injury [167]. Liver fibrogenesis is driven by etiologies that lead to chronic inflammation. Common causes include excessive alcohol intake, viral infection (hepatitis B or C), and non-alcoholic fatty liver disease. Less prevalent causes include autoimmune hepatitis, parasitic infections (schistosomiasis), hemochromatosis, Wilson’s disease, primary biliary cholangitis, and primary sclerosing cholangitis [168]. Cirrhosis is the ultimate end stage of liver fibrosis; it results in over one million deaths annually as the 14th leading cause of death worldwide [169]. Cirrhosis is estimated to affect 1% to 2% of the global population [170]. Significant complications of cirrhosis include failure of liver function, esophageal varices, portal hypertension, hepatic encephalopathy, ascites, spontaneous bacterial peritonitis, hepatorenal syndrome, and hepatocellular carcinoma (HCC) [171].

Experimental models for liver fibrosis are based on in vivo murine models, with common fibrosis-inducing modalities including hepatotoxin administrations (e.g., carbon tetrachloride [CCl4]), thioacetamide (TAA)) that induce acute hepatocellular injury and pericentral liver fibrosis, or bile duct ligation (BDL) to induce cholestasis resulting in periportal liver fibrosis [28,172]. In-vitro models of liver fibrosis have also been utilized through analysis of human hepatocyte cell lines cultured with induced pluripotent stem cells (iPSC-HSCs) [173,174,175]. 

In response to cellular insult, hepatocytes undergo apoptosis and directly modulate the wound repair response through interaction with surrounding cells called hepatic stellate cells (HSCs) [176,177]. HSCs are quiescent vitamin A-storing mesenchymal cells residing in the subendothelial space of Disse that are induced into myofibroblast-like cells in close proximity to apoptotic hepatocytes [178]. 

Genetic lineage studies have identified HSCs as the key ECM-producing cell population in liver parenchymal diseases [179,180,181]. HSCs engulf apoptotic hepatocytes, leading to activation, and upregulation of TGF-β, α-SMA, and collagen 1α1 (COL1A1) [182,183]. Activated HSCs adopt myofibroblast characteristics that enable a contractile, migratory, proliferative, and fibrogenic profile. Activated HSCs secrete copious amounts of ECM components within the space of Disse [184] that can occlude microvascular fenestrations that are crucial to liver functions [185]. Ultimately, pathological levels of ECM deposition result in organ fibrosis and detrimental deterioration of liver function [172,186].

HSCs are recruited from a variety of cell types. Myofibroblast-like cells originate from resident portal fibroblasts (cholestatic cirrhosis) [187], circulating (bone marrow-derived) fibrocytes [39], and epithelial cells that have undergone EMT [185,188]. Endothelial cells can undergo a similar phenotypic transition through endothelial-to-mesenchymal transition (EndoMT) [189]. Mesothelial cells make up around 15% of resident liver cells and also undergo a transition to contribute to myofibroblast populations through MMT [190,191,192]. Lastly, glioma-associated oncogene family zinc finger 1 positive mesenchymal stem cell–like cells also contribute to hepatic myofibroblasts [117,179].

Myofibroblast apoptosis avoidance also influences liver fibrogenesis [193]. Despite the upregulation of extrinsic apoptosis signaling pathway ligands Fas and CD40 [194], activated HSCs adopt an apoptosis-resistant phenotype [195]. HSC persistence ensures sustained HSC activation maintaining the pro-fibrotic environment and perpetuating the fibrotic disease process [196]. Much like apoptotic priming demonstrated in lung fibrosis, HSCs are sensitized to apoptosis but are apoptosis-resistant due to anti-apoptotic signaling. Activated HSCs express a variety of death receptors [197], including TNFR1, FAS, p75 neutrophin (p75NTR) and TRAIL [198]. In CCl4-induced liver fibrosis in rats, exogenous administration of pegylated TRAIL reduced fibrosis and induced cell death of activated HSCs [199]. It was also noted that transgenic mice lacking the p75NTR death receptor showed decreased fibrosis resolution and decreased myofibroblast apoptosis [200]. 

Pro-survival mechanisms enable activated HSCs to persist and avoid apoptosis. Reactive oxidative species-mediated activation of the nuclear-factor kB (NF-κB) signaling pathway [201]. NF-κB survival signaling is activated by TNF and IL-1β [202] and results in upregulation of anti-apoptotic as BCL-XL and BFL-1 [203], TRAF1 and TRAF2 [204], and IAPs, XIAP, FLIP and c-IAP [205]. Pro-survival Akt signaling also decreases HSC apoptosis through Jun N-terminal kinase [206] and FAK activation [207,208]. TGFβ and TIMP1 have also been shown to promote the survival of activated HSCs [178]. PDGF signaling activates HSCs leading to mitochondrial apoptotic priming [209]. These HSCs are dependent on increased BCL-XL expression to avoid apoptosis [210]. This method of myofibroblast apoptosis avoidance is analogous to TGFβ1 mediated survival seen in other fibrotic conditions [11].

Experimental studies have consistently shown hepatocyte apoptosis triggers liver fibrogenesis [211,212] and may be led by hepatocyte p53 over-expression [213]. Kodama et al. 2011 showed that deletion of murine-double-minute 2 homolog (MDM2) in mice, a ubiquitin E3 ligase that targets p53 for degradation, resulted in increased hepatocyte apoptosis, elevated synthesis of profibrotic connective tissue growth factor (CTGF), HSC activation, and resultant liver fibrosis. Furthermore, the removal of p53 abolished this phenotype [214].

p53 plays a role in mediating HSC activation [215]. Experimental studies have shown that reduced p53 expression in HSC leads to excessive liver fibrosis [216] and is thought to be related to reduced p53-dependent HSC senescence [217]. Studies have shown that when HSCs reach their replicative limit, they may adopt a senescent phenotype (pro-inflammatory and anti-fibrogenic) [218,219].

## 8. Renal Fibrosis

Renal fibrosis is an essential component of chronic kidney disease (CKD), an incurable life-threatening pathology [220] that affects nearly 10% of the population worldwide [221]. The incidence of CKD is increasing, resulting in a mounting social and financial public health burden worldwide [222,223]. Etiologies such as hypertension, diabetes, and immune or toxic stimuli cause CKD through chronic inflammation and the development of fibrosis [223,224]. Renal fibrogenesis represents the culmination of a dysregulated wound response following renal injury/chronic inflammation [225,226]. Is it apparent that regardless of the site of renal insult (glomerulus, tubules, or interstitium), the ultimate endpoint of all CKD etiologies is tubulointerstitial fibrosis [227]. Fibrosis plays a major role in CKD progression to end-stage renal disease, currently treated with renal replacement therapy via dialysis or transplantation [224].

Insight into renal fibrosis has been obtained through experimental models such as the common rodent model that utilizes unilateral ureteral obstruction (UUO), whereby one ureter is ligated while the other is a control, leading to fibrosis development within 7 days [228]. Renal tubular injury has been recognized as the most influential site driving renal fibrosis [229,230,231]. In response to injury, renal tubular cells undergo apoptosis, leading to tubular atrophy, reduced kidney function, and associated progression of CKD [232,233,234]. Renal tubular cells can undergo apoptosis through activation of TNF and FAS surface death receptors [235,236] or through activation of the mitochondrial intrinsic signaling pathway via BCL-2 family members proteins BAK and BAX [233,237,238,239]. In both acute and chronic models, inhibition of these proteins reduced renal cell apoptosis [240,241] and suppressed renal interstitial fibrosis [242,243].

Tissue injury will also result in the local activation of inflammatory cells and the release of reactive oxidative species, pro-fibrogenic cytokines, and growth factors that activate myofibroblasts to result in ECM deposition [226,244,245]. Specifically, TGF-β1, interleukins (IL-13, IL-21), and the renin-angiotensin-aldosterone system have been implicated in renal myofibroblast activation [246]. In the setting of non-resolving inflammation/irritants, these processes facilitate renal fibrogenesis [232,247,248]. 

The origins of renal myofibroblasts are disputed but are mainly derived from local interstitial fibroblasts, circulating (bone marrow–derived) fibrocytes, and Gli+ progenitors [249,250,251]. Studies have shown that matrix-producing myofibroblasts are also recruited via partial EMT of tubular epithelial cells [252,253,254]. Other myofibroblast sources include the phenotypic conversion of tubular endothelial cells and differentiation of pericytes [255,256]. In CKD, myofibroblast persistence is thought to play a role in the fibrotic response, however, it is unclear if this persistence is independent or dependent on recurrent local injury/inflammatory stimuli [257]. Furthermore, pro-survival FAK and/or AKT signaling has also been indicated in kidney fibrosis [258,259].

The relationship between p53 and renal fibrosis is complex. Experimental studies have frequently shown tubular cell apoptosis to be linked to renal fibrogenesis [237,260]. A recent study by Liu et al. 2018 demonstrated that tubular cell apoptosis may be related to p53 over-expression [261]. This study examined hypoxia-induced renal fibrosis in a human and rat renal tubular epithelial cells and a mouse UUO model. The results showed p53 to be upregulated, with resultant increased cell-cycle arrest, increased expression of pro-fibrotic cytokines (TGF-β and CTGF), exuberant ECM deposition, and renal fibrosis [261]. Interestingly, inhibition of p53 expression reduced tubular cell apoptosis in rodent models of acute kidney injury (AKI) in the short-term [262] but actually resulted in increased renal fibrosis long-term [263]. 

p53 is thought to play an important role in renal fibroblast activation [264]. Studies have shown renal fibroblasts exhibit marked increased expression of MDM2 in patients with tubulointerstitial fibrosis and UUO mice [249]. However, pre-treatment with Nutlin 3a (p53-MDM2 interaction inhibitor) did not ameliorate fibroblast activation in TIF or UUO [249]. Additionally, studies have shown that in renal myofibroblasts TGF-β1 stimulates p53 phosphorylation which will activate SMAD (small worms and mothers against decapentaplegic) canonical signaling with resultant myofibroblast activation/differentiation, ECM deposition, and finally renal fibrosis [265,266,267].

## 9. Cardiac Fibrosis

Cardiac fibrosis is a significant issue in nearly all etiologies of heart disease [268]. In the heart, parenchymal cells are comprised of muscle cells (cardiomyocytes) rather than epithelial cells [28]. Cardiomyocytes have minimal regenerative capacity and therefore extensive scarring is necessary to prevent cardiac rupture following cardiac injury [40]. 

Cardiac fibrosis can have a detrimental effect on cardiac function [269]. Excessive ECM accumulates within the cardiac interstitium, impairing systolic and diastolic function, and reducing compliance and contractility of the ventricles [270,271,272]. Cardiac fibrosis is categorized into four types based on cause and location: interstitial, replacement, infiltrative interstitial, and endomyocardial [273,274,275,276].

Cardiac fibrosis experimental models include myocardial infarction induction through left anterior descending (LAD) coronary artery occlusion, and pressure-overload-induced cardiac hypertrophy through a transverse aortic constriction (TAC) [28]. Following cardiac insult (e.g., myocardial infarction), cardiomyocyte death is mediated by apoptosis and necrosis [277,278,279]. BCL-2 and BAX have been shown to be expressed in cardiomyocytes [280,281,282]. Cardiomyocyte death stimulates an inflammatory and fibrogenic response to permit cardiac fibroblasts activation and differentiation to myofibroblasts [283,284]. Myofibroblasts are tasked with ECM deposition and wound contraction via α-SMA/periostin [285,286]. This is primarily an adaptive response, but an imbalance in ECM deposition can cause cardiac remodeling, fibrosis, and heart failure [281,287,288,289].

Cardiac myofibroblasts origin is a controversial topic [271]. Genetic lineage studies have identified myofibroblasts to be primarily derived from resident cardiac fibroblasts [117,290], which represent a significant proportion of cardiac cells [291,292]. Other cell types contributing to myofibroblast populations include monocytes/macrophages, endothelial cells, and hematopoietic fibroblast progenitors [293,294,295]. As with other fibrotic conditions transitions to myofibroblasts occur via EndoMT [296,297]. Circulating bone-marrow progenitor cells also contribute to the myofibrocyte population in cardiac injury [298].

Following myocardial infarction, myofibroblast density rapidly increases over a period of weeks [299,300,301,302]. When physiological wound repair is complete, collagen-based ECM becomes organised and subsequently releases mechanical stress, which triggers myofibroblasts to undergo apoptosis [303]. During infarct healing cardiac myofibroblasts have been shown to exhibit the Fas receptor, suggesting extrinsic apoptosis signaling pathway activation [304,305]. However, significant numbers of myofibroblasts will persist for many years which can lead to detrimental cardiac fibrosis [306,307]. This has been noted in pressure overload left ventricular hypertrophy as a form of chronic injury that results in exuberant fibrosis, ventricular wall stiffening, systolic and diastolic dysfunction, and cardiac failure [307,308,309]. 

As noted in other fibrotic conditions, cardiac fibroblast persistence is related to reduced apoptosis and increased proliferation [310,311,312] and is mediated through pro-survival signaling [303]. Cardiac fibroblasts have been shown to be resistant to apoptosis through modulation of the intrinsic signaling pathway one study by Mayorga et al. 2004 demonstrated cardiac fibroblasts exhibit increased expression of BCL-2 to avoid apoptosis, with BCL-2 knockdown resulting in increased cardiac fibroblast apoptosis [313]. Contrary to in other fibrotic conditions, therapeutic strategies have aimed to permit myofibroblast persistence as cardiac fibrosis has an initial cardiac preserving function [40]. Fas/Fas Ligand interaction inhibition in mice 3 days following a myocardial infarction demonstrated decreased myofibroblast apoptosis that resulted in a beneficial thick contractile scar, reduced progression of cardiac dysfunction, and heart failure [304,307].

The relationship between p53 and cardiac fibrosis requires further investigation. In experimental studies, it was shown that mice with p53 knock-out exhibited increased cardiac fibrosis post myocardial-infarction [314,315,316]. Zhu et al. 2013 showed that under-expression of p53 reduced cardiac fibroblast senescence and increased ECM deposition in ischemic myocardium [315]. Experimental studies also demonstrated that reduced expression of p53 is related to pro-fibrotic effects of micro-RNA (miRNA) miR-125b [317] and Sprr2b protein [318] on cardiac fibroblasts. Pang et al. 2021 showed that p53 activation by pharmacological inhibition of RNA polymerase I in cardiac fibroblasts decreased cardiac fibrosis [319]. Together, these results indicate restoring p53 expression in cardiac fibroblasts may decrease their pro-fibrotic tendencies [315].

## 10. Glaucoma

Glaucoma is a chronic-progressive optic neuropathy and a leading cause of irreversible blindness worldwide [320], estimated to affect approximately 76 million people in 2020 [321]. The primary site of glaucoma-related damage is the lamina cribrosa (LC) region of the optic nerve head (ONH) [322], a three-dimensional structure composed of perforated elasto-collagenous cribriform plates [323] that provides structural support to retinal ganglion cell (RGC) axons when leaving the eye to form the optic nerve [324,325,326,327].

Experimental studies have shown that intraocular pressure (IOP) elevation will result in a characteristic ONH cupping [1,328,329] and a thickened, stiffened, and posteriorly displaced LC [330,331,332,333,334,335,336] that exhibits upregulated expression of collagen (I, IV, VI) and elastin [336,337,338,339,340]. Later, LC plates undergo shearing and collapse due to exuberant ECM deposition [323] and subsequent pathological ECM remodeling and fibrosis [327]. This process culminates in a thin, fibrotic, architecturally altered LC [320,323,327,330,331,336,341] that obstructs retinal ganglion cell (RGC) axon axoplasmic flow [320,327,329,342,343] and leads to progressive degeneration of RGC axons and associated irreversible vision loss [344,345,346,347,348,349,350].

Previous work by our research group suggests that resident glial fibrillary acid protein (GFAP) negative LC cells play a crucial role in ECM remodeling and fibrosis at the ONH in glaucoma [322,351,352]. Significantly, LC cells bear similarities to myofibroblasts responsible for fibrotic disease development [13,353]. These similarities include the expression of α-SMA, COL1A1, elastin, and fibronectin, as well as bone morphogenic proteins (BMPs) [354,355]. Furthermore, LC cells exposed to cyclic mechanical stretch [352,356,357,358,359], oxidative stress [360], hypoxia (ONH ischemia) [361] and TGF-β1 [356] adopt a profibrotic response state that results in upregulated ECM gene expression [352,356,362,363].

The role p53 and myofibroblast apoptosis plays in the cells within the glaucomatous ONH has not been categorized. Therefore, deciphering the apoptotic signature of LC cells and elucidating its role in glaucomatous fibrogenesis could generate a greater comprehension of the mechanisms driving fibrotic glaucomatous ONH remodeling. This represents a novel area of investigation in glaucoma research and could lead to novel therapeutic interventions.

## 11. Therapeutics

Under normal conditions, p53 levels are decreased through inactivation and negative regulation by the oncoprotein MDM2 [75,364,365]. MDM2 is an E3 ubiquitin ligase that uses the ubiquitin-proteasome system (UPS) to target p53 for degradation [366]. MDM2 also inactivates p53 through nuclear exportation [73,74,364,367,368] and through direct binding to inhibit p53’s transcriptional activity [369,370].

A greater understanding of the p53-MDM2 interaction has enabled the emergence of novel therapeutics that aim to inhibit MDM2 binding to p53 and thus ensure p53 stabilization and activation [371]. These synthetic compounds work by preventing the transactivation domain of p53 binding to a deep hydrophobic pocket on MDM2, specifically targeting three amino acid residues (Phe19, Trp23 and Leu26) in p53 that are primarily responsible for this protein-protein interaction [372,373]. Amongst the earliest investigated therapies were the imidazoline derivatives (better known as nutlins), especially nutlin 3a [374]. Nutlin 3a mimics the aforementioned three amino acid residues in p53 necessary for MDM2 interaction, acting as a competitive inhibitor of p53 binding to MDM2 [364]. Preclinical studies showed that nutlin 3a increased p53 concentrations, enhanced apoptosis, and decreased tumorigenicity in p53 cancer cells [373]. Nutlin-derived small molecule MDM2 inhibitors phase I clinical trials have shown administration of the potent, orally bioavailable nutlin imidazoline compound RG-7112 (Roche Pharmaceuticals) results in activation of p53, p21, and induction of apoptosis in human tumours [374,375]. However, due to high dosing levels, patients experienced notable gastrointestinal toxicity, neutropenia, and thrombocytopenia [376,377].

More potent MDM2 inhibitors such as the pyrrolidine idasanutlin (formerly RG7388) have been developed to address this side-effect profile [378]. Recently, phase III MIRROS (MDM2 antagonist Idasanutlin in Relapsed or Refractory acute myeloid leukemia [AML] for Overall Survival) trial evaluated the efficacy and safety of the small-molecule MDM2 antagonist idasanutlin plus cytarabine in patients with relapsed/refractory AML and demonstrated that nutlin treatment has no effect on overall survival or complete remission rates [379]. Again, nutlin dose-related toxicities and adverse side effects were an ongoing concern [380].

Research into therapeutic avenues in the area has aimed at also targeting MDMX (also known as MDM4) [381]. MDMX possesses a high degree of homology to MDM2, especially in its N-terminal p53 binding domain. In combination with MDM2, MDMX contributes to maintaining low levels of p53 in the normal cell by directly binding and inhibiting the transactivation domain of p53 [382]. MDMX itself cannot induce p53 degradation but modulates MDM2’s E3 ligase activity by heterodimerization to MDM2’s RING finger domain at its C-terminus [383]. MDM2 and MDMX can work independently, or alternatively, form a complex that is more effective at inhibiting p53 transactivation or enhancing p53 turnover [383]. Additionally, when stimulated by DNA damage, MDM2 can directly ubiquitinate and degrade MDMX upon DNA damage stimuli [384]. Increased expression of MDMX is noted in cancer and promotes tumorigenesis [385].

The development of MDM2-MDMX dual inhibitors has been challenging to date. The most promising clinical trials have studied ALRN- 6924 (Aileron Therapeutics) [386]. This is a cell-penetrating α-helical peptide [387]. that showed increased survival rates in an AML xenograft model [388]. This agent is being evaluated in phase I/II clinical trials in wild-type p53 haematological and solid malignancies and is reported to have an improved adverse side-effect profile [389]. These clinical trials to date highlight the need for further research in the field of targeted MDM2 and MDM2-MDMX small molecule therapeutics. 

Outside of oncology, MDM2 has been identified to have a role in cardiovascular disease and heart failure [390], diabetes [391], neurodegenerative diseases [392], nephropathy [393], obesity [394], and autoimmune and inflammatory conditions [395]. Small molecule MDM2-inhibitors are being utilized as therapeutics in pre-clinical studies in systemic lupus erythematosus [396] and crescentic glomerulonephritis [397] with provisionally promising results. 

Nutlins have also been trialed in in vivo experimental studies in pulmonary [71,163], liver [214], cardiac [398,399], and renal fibrotic models [249,400] with the resultant restoration of p53 expression and amelioration of fibrosis. Further clinical trials and a greater understanding of p53 and myofibroblast apoptosis-resistance in organ fibrosis are required to develop future novel therapeutic interventions.

## 12. Conclusions

Organ fibrosis results in progressive disruption of normal tissue architecture leading to detrimental deterioration in physiological function, and significant morbidity/mortality [13,20,36]. This review highlights the essential role myofibroblasts play in fibrotic disease development throughout the human body [13,353]. In organ fibrosis, myofibroblasts have been shown to adopt an apoptotic-resistant phenotype to perpetuate fibrosis [2,11] and myofibroblast persistence leads to myofibroblast expansion, ECM deposition and remodeling, persistent tissue tension/contraction, and the formation of a pathological scar [106].

Apoptosis can be initiated by p53, a potent transcription factor that is activated in response to diverse stresses and environmental insults [74,86]. p53 is also responsible for the induction of cell-cycle arrest and senescence [75] to prevent the emergence of transformed cells with genetic instabilities [76]. p53 potentiates apoptosis signaling through transcription-dependent means that stimulate genes involved in the extrinsic pathway, intrinsic pathway, and execution factors [84,88]. p53 also stimulates apoptosis by transcription-independent means through interaction with BCL-2 family member proteins to directly induce MOMP [84,89,90,91,92]. In most, if not all, human cancers, inactivation of p53 disrupts its ability to suppress carcinogenesis, thus transforming the “Guardian of the Genome” into a “Rebel Angel” [96]. 

This review shows that p53 dysregulation has been consistently linked to fibrotic disease development. In all types of organ fibrosis, initial tissue injury or irritant will result in apoptosis of the resident cell group. The evidence shows that this apoptotic signaling is initiated/sustained by the over-expression of p53. Resident cells undergoing apoptosis will then initiate a wound repair response system that will directly activate local or recruited fibroblasts that will differentiate into specialized apoptotic-resistant myofibroblasts. The role of p53 in myofibroblast persistence is unclear, however, recent experimental studies have demonstrated evidence of p53 under-expression in fibroblasts in organ fibrosis, although additional research is needed to further elucidate this relationship.

## Figures and Tables

**Figure 1 ijms-24-06737-f001:**
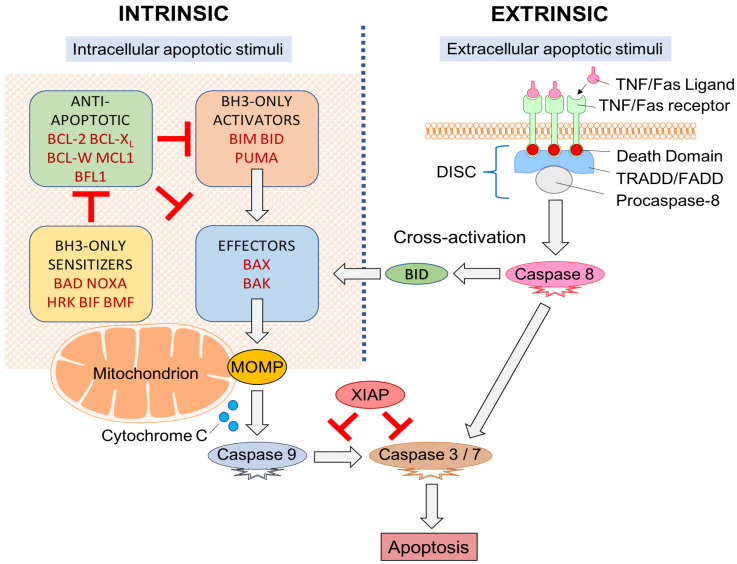
Apoptotic signaling pathway. “Extrinsic” and “intrinsic” pathways both activate caspases. Cross-activation between pathways is made possible through BID (a BCL-2 family member protein). BCL-2 proteins act within the mitochondrion with the relative expression of pro- and anti-apoptotic proteins determining if the apoptotic threshold is crossed and MOMP can proceed.

**Figure 2 ijms-24-06737-f002:**
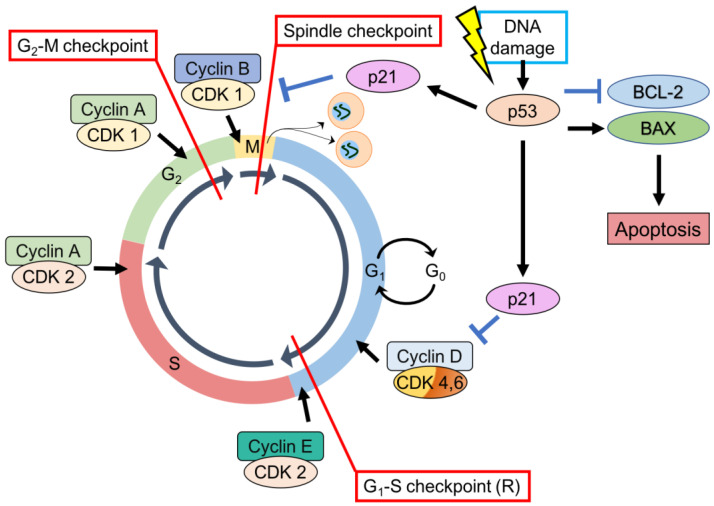
p53, CDK substrates, CDK inhibitors, and cyclins—regulation of the cell-cycle. (1) G1-S checkpoint. (2) G2-M checkpoint. (3) Spindle checkpoint. p53 can also stimulate intrinsic apoptotic signaling pathways through stimulation of BAX expression and inhibition of BCL-2 expression.

## Data Availability

Not applicable.

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
