# Peer review of "p53 and Myofibroblast Apoptosis in Organ Fibrosis"

_ijms, 2023, doi:10.3390/ijms24076737_

Round 1

Reviewer 1 Report

In this review ‘p53 and Myofibroblast Apoptosis in Organ Fibrosis’ authors discuss the role of p53 dysregulation in myofibroblast apoptosis to the development of novel therapeutic.

I would like to see a revision to emphasize on Therapeutics section, although authors discuss about inhibitors to target direct binding of p53-MDM2. While Nutlin 3a is promising to activate p53-dependent apoptosis but it’s derivatives have the toxic side effects in patients due to high doses.

Authors should include more clinical studies focusing to understand the underneath mechanism of p53 to overcome myofibroblast apoptosis-resistance in organ fibrosis.

SNP at codon 72 of p53 affects the IFN-dependent innate immune response against virus infection (published in International Journal of Molecular Sciences 22 (16), 8660      13        2021 and Mol. Cell. Biol. 2011, 31, 1201–1213).

I would like to see in revised version to discuss whether SNP at codon 72 of p53:

1. has its effect on activation of apoptosis through cell cycle regulation and

2. therapeutic response myofibroblast apoptosis-resistance in organ fibrosis by inhibitors.

2. effect on binding to MDM2.

3. development of myofibroblast apoptosis-resistance in organ fibrosis.

Authors should widen their area to in vivo or patients related studies especially myofibroblast apoptosis-resistance in organ fibrosis.

Overall, the Study is comprehensive, well-designed, and has an acceptable flow.

Author Response

Many thanks for your comments regarding our paper, we have researched SNP codon 72 of p53 and have added the following paragraph with references to the article in the p53 section. Furthermore, we have elaborated on the therapeutics section to include more information on clinical trials surround p53 targeted therapies. Regarding the request for further clinical studies in the area of myofibroblast apoptosis-resistance in organ fibrosis, this is not possible as clinical/patient based studies/trials in this area do not exist yet. This is one of the main reasons we chose to write this review article – we aimed to bring a cohesion to a broad and disjointed field of laboratory studies that may help progress the field further and hopefully aid future clinical studies in the area.

One of the regions of p53 that is believed to be required for its ability to induce apoptosis is its proline-rich domain [93]. Within this domain, there is a common single-nucleotide polymorphism (SNP) at codon 72 encoding either an arginine (R72) or a proline (P72) residue [93]. Interestingly, this variance in polymorphic form results in a marked alter-ation in the biochemical structure and function of p53 [94]. In-vitro studies have shown that p53 with expressions of the P72 allele demonstrate an increased ability to induce se-nescence and cell-cycle arrest [95] through increased transactivation of p21/Waf-1, which blocks CDK activity leading to growth arrest at the G1 phase of the cell-cycle [96]. p53 with expressions of the R72 allele exhibit an increased ability to induce apoptosis [97] thought to be related to elevated mitochondrial localization and stimulation of pro-apoptotic BCL-2 family member protein BAK [98].        

Reviewer 2 Report

In this article, Kealan McElhinney and co-workers reviewed the literature on organ fibrosis, showing that a common feature of this disorder is myofibroblasts adopting an apoptotic-resistant and highly proliferative phenotype; this  perpetuates fibrosis while leading to myofibroblast expansion with  major deposition/remodelling of the extracellular matrix.

The authors first described the physiological wound repair and explained how its dysregulation may lead to fibrosis through the activation and accumulation of myofibroblasts. After a detailed  description of the molecular mechanisms driving apoptosis and of the role of p53 in cell cycle arrest, they summarized the current knowledge  on apoptotic failure and p53 dysregulation in the etiology of organ fibrosis, with special attention to  the fibrosis occurring in lung, liver, kidney, heart  and in glaucoma.  p53 has been consistently linked to the development of the fibrotic disease.

This is an interesting and exhaustive review which may surely be of interest for the scientists and clinicians in the field.

I have only minor concerns on purely formal details.

·     -    The abbreviations should be explained the first time they are used only (see, e.g. lines 24, 387 and 566; lines 270, 393 and 469; lines 120 and 305; lines 273 and 398; lines 396 and 521)

·      -   The authors used a number of abbreviations, some of which are actually useless, as used a few times or even once only (e.g., line 329, FSSCP; Line 461, ROS)

·    -     Linen 343: “The role between p53 expression in pulmonary fibroblasts was extensively re-343 searched by Nagaraja et al.....”: the role between p53 and what?

·      -   Lines 380 1ne 387: “Disse” not “Dissé”

·    -     Line 581: Which is the difference between “biomechanical and biomechanical fibrotic mechanisms”?

·     -    The whole text should be checked for typing errors (e.g., see lines 262,  364 and 372)

·     -   These articles might also be considered:

Yu S, Ji G, Zhang L. The role of p53 in liver fibrosis. Front Pharmacol. 2022 Oct 24;13:1057829. doi: 10.3389/fphar.2022.1057829

Overstreet JM, Gifford CC, Tang J, Higgins PJ, Samarakoon R. Emerging role of tumor suppressor p53 in acute and chronic kidney diseases. Cell Mol Life Sci. 2022 Aug 9;79(9):474.

Author Response

Many thanks to this reviewer for their suggestions and kind comments, please find a point by point response to the various aspects of their reviewer comments

  •    -    The abbreviations should be explained the first time they are used only (see, e.g. lines 24, 387 and 566; lines 270, 393 and 469; lines 120 and 305; lines 273 and 398; lines 396 and 521)

Addressed with thanks

  •     -   The authors used a number of abbreviations, some of which are actually useless, as used a few times or even once only (e.g., line 329, FSSCP; Line 461, ROS)

Addressed with thanks

  •   -     Linen 343: “The role between p53 expression in pulmonary fibroblasts was extensively re-343 searched by Nagaraja et al.....”: the role between p53 and what?

Addressed with thanks

  •     -   Lines 380 1ne 387: “Disse” not “Dissé”

Addressed with thanks

  •   -     Line 581: Which is the difference between “biomechanical and biomechanical fibrotic mechanisms”?

Addressed with thanks

  •    -    The whole text should be checked for typing errors (e.g., see lines 262,  364 and 372)

Addressed with thanks

  •    -   These articles might also be considered:

Thank you for the following articles.

Yu S, Ji G, Zhang L. The role of p53 in liver fibrosis. Front Pharmacol. 2022 Oct 24;13:1057829. doi: 10.3389/fphar.2022.1057829

Thank you - this article was very helpful in finding out more on p53 role in HSC activation

Overstreet JM, Gifford CC, Tang J, Higgins PJ, Samarakoon R. Emerging role of tumor suppressor p53 in acute and chronic kidney diseases. Cell Mol Life Sci. 2022 Aug 9;79(9):474.

Thank you - this article was very helpful in finding out more on p53 role in renal fibroblast activation